The effect of cycling hypoxia on MCF-7 cancer stem cells and the impact of their microenvironment on angiogenesis using human umbilical vein endothelial cells (HUVECs) as a model

Alhawarat Fuad M. 1
Hammad Hana M. 1
Hijjawi Majd S. 2
Sharab Ahmad S. 2
Abuarqoub Duaa A. 1
Al Shhab Mohammad A. 2
Zihlif Malek A. m.zihlif@ju.edu.jo 2
1 Department of Biological Sciences, School of Science, The University of Jordan , Amman , Jordan
2 Department of Pharmacology, School of Medicine, The University of Jordan , Amman , Jordan
Coates Philip
Electronic publication date: 2019 Jan 8
Publication date: 2019
Volume: 7
Electronic Location ID: e5990
Received 2018 Jun 7; Accepted 2018 Oct 22
Copyright: ©2019 Alhawarat et al.
Copyright year: 2019
Copyright holder: Alhawarat et al.
License: This is an open access article distributed under the terms of the Creative Commons Attribution License, which permits unrestricted use, distribution, reproduction and adaptation in any medium and for any purpose provided that it is properly attributed. For attribution, the original author(s), title, publication source (PeerJ) and either DOI or URL of the article must be cited.
License URL: https://creativecommons.org/licenses/by/4.0/

Keywords: Cancer Stem Cell, Hypoxia, Angiogenesis, Chemoresistance, Breast Cancer, Mammosphere, CSCs microenvironment

Funding: Deanship of Scientific Research, University of Jordan, Amman Jordan This work was supported by the Deanship of Scientific Research, University of Jordan, Amman Jordan. The funders had no role in study design, data collection and analysis, decision to publish, or preparation of the manuscript.

==============================
Background

Breast cancer is the most common type of cancer among females. Hypoxia mediates cancer hallmarks and results from reduced oxygen level due to irregularities in tumor vascularization or when the tumor size prevents oxygen diffusion and triggers angiogenesis to compensate for low oxygen. Cancer stem cells (CSCs) are a rare subpopulation, able to self-renew and to give rise to tumor-initiating cells. It is proposed that CSCs’ secretions help to recruit endothelial cells via angiogenic factors to establish tumor vascularization. In the tumor microenvironment, the effect of hypoxia on CSCs and the impact of their secretions on triggering angiogenesis and tumor vascularization remain questionable. In this study, three-dimensional (3D) CSCs derived from MCF-7 were directly exposed to repetitive long-term cycles of hypoxia to assess its effect on CSCs and then to evaluate the role of the hypoxic CSCs’ (CSCsHYP) secretions in angiogenesis using (HUVECs) as a model for tumor neovascularization response.

Methods

CSCs derived from MCF-7 cell-line were expanded under repetitive, strictly optimized, long-term/continuous and intermittent hypoxic shots for almost four months to assess hypoxic effect on CSCs, sorted based on CD44+/CD24− biomarkers. Hypoxic phenotype of CSCsHYP was evaluated by assessing the acquired chemoresistance using MTT assay and elevated stemness properties were assessed by flow cytometry. To evaluate the effect of the secretions from CSCsHYP on angiogenesis, HUVECs were exposed to CSCsHYP conditioned-medium (CdM)—in which CSCs had been previously grown—to mimic the tumor microenvironment and to assess the effect of the secretions from CSCsHYP on the HUVECs’ capability of tube formation, migration and wound healing. Additionally, co-culture of CSCsHYP with HUVECs was performed.

Results

CSCsHYP acquired higher chemoresistance, increased stemness properties and obtained greater propagation, migration, and wound healing capacities, when compared to CSCs in normoxic condition (CSCsNOR). HUVECs’ tube formation and migration abilities were mediated by hypoxic (CSCs) conditioned media (CdM).

Discussion

This study demonstrates that chemoresistant and migrational properties of CSCs are enhanced under hypoxia to a certain extent. The microenvironment of CSCsHYP contributes to tumor angiogenesis and migration. Hypoxia is a key player in tumor angiogenesis mediated by CSCs.

Introduction

Cancer stem cells (CSCs) are a subpopulation in cancer tissue that slowly divide with an ability to regenerate by uneven cell division (Huang & Rofstad, 2017), contributing to cancer relapse, invasiveness and thus higher cancer mortality (Peitzsch et al., 2017). CSCs have the same intrinsic features as normal stem cell populations found throughout the body, including the ability of self-renewal and differentiation (Batlle & Clevers, 2017).

In most solid tumors, including breast carcinoma, hypoxia is a mutual hallmark that is proven to associate with poor prognosis of cancer patients (Muz et al., 2015). Various stress-response pathways that drive cancer cells toward a self-stabilizing and anti-apoptotic phenotype are activated upon the exposure of cells to sub-physiologic concentrations of oxygen (≤1% O2) or hypoxia (Crowder et al., 2014). Hypoxia stabilizes the hypoxia-inducible factor-α (HIFα) proteins that contribute to various pro-survival processes in cancer cells, such as angiogenesis, abnormal proliferation and metabolic alterations (Bristow & Hill, 2008; Kietzmann, Mennerich & Dimova, 2016; Kong et al., 2014), and thus increases the expression of pro-survival factors that support radioresistance and chemoresistance (Crowder et al., 2014; Senthebane et al., 2017). The exact signaling pathways by which hypoxia activates its effects are complex and remain under investigation. Among well-recognized mechanisms are elevated genomic instability and irregular cell division (Hammer et al., 2007), uncontrolled reactive oxygen species and redox mechanisms, aerobic glycolysis metabolic shift (Denko, 2008), and reduced expression caused by proapoptotic-factors (Gordan et al., 2008).

Cycling hypoxia determines the state of oxygen level and its dispersal in solid tumor tissues. Dysfunctional heterogenic blood supply and tumor irregular vascularity in cancer cause oxygen fluctuation for irregular periods with intermittent re-oxygenation intervals (Brurberg et al., 2004; Muz et al., 2015). Acute hypoxia is an immediate brief exposure to a short-term hypoxic status where the blood vessel occlusion lasts for at least few minutes and up to 72 h (Chaplin, Olive & Durand, 1987; Muz et al., 2015). These changes in the bloodstream and the limited oxygen availability result in chronic hypoxia, which is more apparent in large-sized tumors and energizes long-term cellular changes (Eales, Hollinshead & Tennant, 2016). In experimental protocols, chronic hypoxic exposure is obtained by incubating cells in hypoxia up to several weeks (Cassavaugh & Lounsbury, 2011). Indeed, clinical responses to cancer therapy are directly affected by both chronic and acute hypoxic sections in solid tumors, inducing their growth, metastatic ability, and cell death resistance (Al-Hajj et al., 2003).

Angiogenesis, which is defined as tumor neovascularization is essential for tumor development and maintenance (Kaur et al., 2005; Zhao & Adjei, 2015). It is also considered an important cellular parameter of tumorigenesis (Wang et al., 2015). Elevated levels of vascular endothelial growth factors (VEGFs) are expressed by endothelial cells involved in angiogenesis and bone marrow-derived progenitor cells, and moreover, many types of human cancer cells can also secrete VEGFs (Kerbel, 2008). Evidence from different studies indicates that CSCs show greater potential for tumor initiation, generation, and production of higher levels of VEGFs than the non-stem cells in cancer population, thus exhibiting more potent proangiogenic capability (Ping & Bian, 2011; Wang et al., 2016). CSCs with elevated VEGF levels are typical example for the significance of angiogenesis in tumor progression (Ojha et al., 2017).

In conclusion, CSCs’ secretions help to recruit endothelial cells via angiogenic factors to establish tumor vascularization. Moreover, hypoxia triggers angiogenesis in cancer tissues. The studies that investigate the effect of long-term hypoxia on CSCs are extremely rare, so the cross-talk between long-term hypoxia and its effect on CSCs and their role in angiogenesis remains questionable. In this context, we hypothesized that repetitive exposure to cycles of hypoxia/normoxia conditions may be a driving force that promotes a highly resistant and invasive CSCs subpopulation, and CSCs are proposed to generate higher levels of tumor vascularization, due to induced angiogenesis, than regular cancer tissues.

Accordingly, the present study aims at investigating the effect of long-term hypoxia on CSCs and the impact of CSCsHY P secretions on angiogenesis in the tumor microenvironment. CSCs were derived from the cell-line and were expanded under repetitive strictly optimized long-term and intermittent hypoxic shots to assess the effect of hypoxia on them. Both intermittent and long-term hypoxia were directly applied for almost four months on sorted and identified CSCs as 3D mammospheres. Thereafter, to assess the impact of secretions of CSCsHY P on angiogenesis, we used the conditioned-medium (CdM) in which CSCsHY P were expanded and grown, as it comprises the CSCHY P secretions, to treat HUVECs and then examine their angiogenic ability.

Materials and Methods

Workflow for the methodology adopted in this study is summarized in (Fig. 1).

Figure 1 Summary of methodology workflow adopted in this project.

The first stage was to establish CSCs sub-population from parental MCF-7 cells by several steps of culturing cells in ultra-low serum-free media and sorting to enrich stemness biomarker that was finally identified by flow cytometry after 21 days. Stage two included exposing CSCs to hypoxic and normoxic conditions in optimized intervals for over than four months and collecting conditioned starvation media in which they were grown. Also, multiple assays were performed using hypoxic and normoxic CSCs to test the hypoxia effect on CSCs. Stage three was performed to assess the effect of CSCs secretions in the angiogenesis.

Culture conditions

The breast cancer cell line (MCF-7) was purchased from the American Type Culture Collection (ATCC, USA), and was expanded as a monolayer in vented 75 cm2 cell culture flasks (Membrane Solutions, USA) using RPMI 1640 media (HyClone, Logan, UT, USA). The media was supplemented with heat-inactivated fetal bovine serum (FBS) 10% (v/v) (HyClone, USA), antibiotics (100 µg/ml streptomycin and 100U/ml penicillin) (HyClone, USA), and 1% 2mM L-glutamine (HyClone, Logan, UT, USA). The cells were incubated in a designated incubator (NuAire, Shanghai, China) at (5% CO2 at 37 °C). All cell culture procedures were performed in sterile conditions under a class II biological safety cabinet (Heal-Force, Shanghai, China). All used materials and disposables were disinfected with ethanol 76% before use.

Mammosphere cultures

To obtain CSCs from MCF-7 cell line and propagate them as mammospheres, MCF-7 cells were expanded in complete growth media described above as a single-cell suspension in a standing flask to enhance the growth of CSCs as they tend to grow in non-adherent conditions. Two days later, remaining cells were collected by centrifugation, washed in phosphate buffer saline, and seeded in six well ultra-low adherence plates (Corning, Corning, NY, USA) at a density of 10,000 cells/well. Cells were expanded in serum-free DMEM/F12 (Sigma-Aldrich) media, supplemented with 20 ng/mL basic fibroblast growth factor (GIBCO, USA), 20 ng/mL epidermal growth factor (MPBio, Santa Ana, CA, USA), 2% B27 (GIBCO, USA), 10 µg/mL insulin (MPBio, Santa Ana, CA, USA), 0.5 µg/mL hydrocortisone (Nacalai, Japan), 0.4% bovine serum albumin (Sigma, St. Louis, MO, USA), and 2 mM L-Glutamine (Biowest Co., Nuaillé, France). This is called mammospheres culture media. Cells were preserved in a humidified incubator at 37 °C, 5% CO2. Every two days, fresh media (2 mL) were added to each well (without removal of the old one). At day seven, primary mammosphere clusters were counted, then collected and centrifuged to form a pellet. In order to obtain single cells, the pellet was enzymatically disaggregated with trypsin and dispersed by passing through a 40 µm pore-size filter (Millipore, Billerica, MA, USA). Single cells of primary mammospheres were collected for sorting, using beads technique and identified by flow cytometry (detailed below) before and after sorting. Disaggregated cells were cultured in ultra-low adherence flask (Corning, Corning, NY, USA) instead of well plates to obtain secondary mammospheres. Then, the same procedure was applied again to obtain tertiary and quaternary mammospheres.

To calculate the percentage of mammosphere forming efficiency (%MFE), the number of mammospheres was divided by the number of seeded cells and multiplied by hundred. Mammospheres were counted under 10x magnification of an inverted light microscope (Olympus, Tokyo, Japan). Twenty one (21) days after initial seeding in ultra-low attachment and serum-free media, a sufficient mammospheres’ population was established.

Cell viability

Cellular viability was evaluated using the trypan blue method. After de-attachment and good homogenization of cells until there was a single-cell suspension, a sample was taken and diluted in 1:1 ratio with trypan blue (Sigma-Aldrich, Gillingham, UK). Then 10 uL of the diluted mixture was loaded on a hemocytometer (Neubauer Double, Zuzi, Spain) and inspected under an inverted light microscope (Olympus, Tokyo, Japan). Cells with shiny white appearance were considered viable and counted because their cell membrane was impermeable to trypan blue, while blue-colored cells were disregarded. Hypoxic cells’ viability was checked upon every other split and compared to its normoxic counterparts to assess proliferation changes. While mammosphere cells were enzymatically disaggregated into single-cells in order to observe viability (Cassavaugh & Lounsbury, 2011).

CSCs sorting with magnetic beads

CD44+/CD24− are the expression biomarkers of interest in MCF-7 derived mammospheres. At day seven of the initial culture of mammospheres in 6-well plate, CSCs enriched with CD44+/CD24− were sorted from MCF-7 mammospheres using MagCellect CD24−/CD44+ Breast Cancer Stem Cell Isolation Kit (R&D System, Cat. MAGH111) according to their protocol. Initially, CD24+ cells were labeled and removed magnetically. Then from the CD24− population, CD44+ cells were sorted magnetically using a biotinylated human antibody and streptavidin-conjugated magnetic beads as a positive selection model. The efficiency of sorting was assessed by staining recovered cells with fluorochrome-conjugated anti-human CD44+ and CD24−antibodies. Isolated CD44+/CD24− cells were later identified via FACS analysis.

Exposure to hypoxia

To create hypoxic conditions, an anaerobic atmosphere generating system, AnaeroGen Compact (Oxoid, Basingstroke, UK) was used. The AnaeroGen system is used in microbiology area to create the low oxygen conditions needed for the growth of anaerobic bacteria. Here, we adopted the AnaeroGen system to induce hypoxia into CSCs. The system consists of a gas generatingsachets and tightly sealable bags. The sachet reacts promptly upon contact with air and consumes oxygen thus reducing oxygen concentration to less than 1% inside the bag. It has been confirmed in previous studies by using biochemical and electronic testing that this technique reduces oxygen content in plastic bag to below 1% within 30 min (Chenevier-Gobeaux et al., 2013; Mellor et al., 2005). The vented ultra-low attachment 75 cm2 tissue culture flasks were placed inside the plastic bags and the sachets were placed inside then sealed. After 21 days of culturing the mammosphere CSCs, they were divided into two groups and exposed to hypoxic conditions intermittently (INTR.) or continuously (CONT). CSCs in the intermittent group were exposed to 10, 20, 30 or 40 hypoxic shots for 8 h three times a week, denoted as (INTR.10, INTR.20, INTR.30, and INTR.40) respectively, while CSCs in the continuous group was exposed to 5, 10, or 15 hypoxic shots for 72 h once per week, denoted as (CONT.5, CONT.10, and CONT.15). Apart from these optimized hypoxic shots, CSCs mammospheres were also incubated in as control, using the same culture media and conditions that were used for expansion of mammospheres detailed above.

Flow cytometry to examine enrichment of CSCs mammosphere with CD44+/CD24− under Hypoxia/Normoxia

Flow cytometry was used to identify and characterize CD44+/CD24− surface phenotype, which is in direct proportion to stemness, and was performed for parental MCF-7 cells, un-sorted mammospheres, sorted CSCs mammospheres three days after sorting, CSCs mammospheres which were 21 days-old all under normoxic conditions and finally for CSCsHY P exposed to shots of INTR.20, INTR.40, CONT.5 and CONT.15. Mammospheres were dispersed to obtain single-cell populations as described above. The CSCs’ mammosphere pellet was washed in phosphate-buffer saline (PBS) (HyClone, USA) with 2% bovine serum albumin and stained with APC-anti-mouse CD24 and PE-anti-mouse CD44 antibodies (BD Pharmingen, CA). CSCs mammospheres were incubated in ice for 30 min, washed twice with PBS and then fixed in PBS containing paraformaldehyde. The flow cytometric analysis was performed on a BD FACSCalibur system (BD Biosciences, San Jose, CA, USA), and the identification was performed with the BD Cellquest software (BD Biosciences).

Cytotoxicity evaluation using MTT assay

To examine the chemoresistance of MCF-7 parental cells and their derived CSCs, both CSCsNOR and CSCsHY P, drug resistance towards doxorubicin was used as an indicator. A CellTiter Cell Proliferation Assay Kit® (Promega, Madison, WI, USA) was used. This assay includes 3-(4,5-Dimethylthiazol-2-yl)-2,5-diphenyl tetrazolium bromide (MTT) which is a yellow tetrazole reduced by viable cells to purple formazan. Consequently, surviving cells after being exposed to multiple concentrations of doxorubicin can be quantified by spectrophotometry. The MTT proliferation assay was performed on MCF-7 parental cells seeded as a monolayer in an adherent 96-well plate. Moreover, CSCsHY P were dissociated into single-cells, filtered, counted then seeded at a 7 × 103 cells/well density in a 96-well plate and incubated for 24 h before treatment. Each concentration was tested in triplicate on CSCs exposed to (INTR.10, INTR.20, INTR.30, and INTR.40) or exposed to (CONT.5, CONT.10, and CONT.15). A control experiment was done for sorted and identified CSCsNOR from mammospheres cultured under normoxic conditions.

Incremental serial dilutions of doxorubicin concentrations from 0.01 µM to 200 µM were used. After 72 h of incubation, old media was aspirated, and 100 µl of fresh media was added to each well. Then, 15 µl of MTT reagent was added to each well and incubated for 4 h. Solubilization/stop solution was added to each well and incubated for an hour. Absorbancy was read using an Elisa reader (Sunrise basic sciences, Austria) at 570 nm wavelength. The half maximal inhibitory concentration (IC50) was calculated using GraphPad PRISM®7.04 software (GraphPad Software, Inc.).

Endothelial cells isolation from umbilical cords

Human umbilical cords were obtained from two delivering females at the Jordan University Hospital after getting the approval of their Institutional Review Board (IRB), decision number 98/2018 and approval number (67/2018/481) and both patients having signed an informed consent before sample collection.

Enzyme digestion technique was used to isolate endothelial cells from the human umbilical cord. Debris on the outer surface of the cord was wiped using antiseptic solution. Any clots formed inside the vein were removed by slight squeezing. The umbilical vein was washed with RPMI medium (HyClone, Logan, UT, USA) containing 1% penicillin/streptomycin (HyClone, Logan, UT, USA) to remove excess blood and debris using 20 mL syringe inserted into the cord. Then, 0.1% of type I collagenase (Sigma-Aldrich, St. Louis, MO, USA) was put inside the vein where it was incubated for 20 min at 37 °C for digestion. Hereafter, the vein was washed with 20 mL RPMI media (HyClone, Logan, UT, USA) to stop collagenase proteolytic activity and was centrifuged at 1,500 rpm for 5 min. The resultant pellet was resuspended in 5 ml endothelial cell growth medium EGM-2 media (Lonza, Walkersville, MD, USA) which consists of the endothelial cell basal media EBM-2 with 10% FBS and other additives (including VEGF, bFGF- Fibroblast Growth Factor, EGF-Epidermal Growth Factor, and IGF-Insulin Growth Factor). HUVECs were expanded on 0.2% gelatin-coated T25 tissue culture flask and incubated overnight. Next day media was renewed to get rid of cells debris. Only passages earlier than six of HUVECs were used in all experiments.

Collection of conditioned-medium from hypoxic/normoxic CSCs

In order to obtain the CdM, CSCs were seeded as single cells at a density of 10,000 cells/cm2 in an ultra-low adherence 75 cm2 flasks with serum-free DMEM/F-12 and with supplements (mammospheres culture media detailed above). CSCs mammospheres were starved overnight by washing them twice with serum and supplement free DMEM/F-12 media, then centrifuged at 300 rpm for 10 min. Finally, the media was renewed without serum and without supplements for 8 h for intermittent and 72 h for the continuous hypoxic condition. The resultant CdM from where CSCs grown under hypoxia or normoxia were collected, centrifuged at 300 rpm for 10 min to remove cellular components and filtered through 0.2 µm pore-size filters (Millipore, Billerica, MA, USA). CdM from flasks under normoxic conditions were used as a control. Aliquots of the CdM were stored at −80 °C before being used. CdM were collected from eight sources, from CSCsNOR, and from CSCsHY P grown under the hypoxic shots (INTR.10, INTR.20, INTR.30, INTR.40, CONT.5, CONT.10, or CONT.15) and were utilized in successive experiments as discussed subsequently.

Capillary-like tube structure formation assay using CSCs’ mammospheres (CdM) on HUVECs

The effect of the factors secreted in the CdM obtained from CSCsHY P and from CSCsNOR on the angiogenic ability of HUVECs was examined through the formation of capillary tube-like structures by HUVECs, using the in vitro matrigel angiogenesis assay to mimic the microenvironment of CSCs. To eliminate the impact of growth factors in Matrigel, reduced growth factor Matrigel (BD Biosciences, San Jose, CA, USA) was used. HUVECs were incubated for 6 h in endothelial basal medium (EBM-2, Lonza) without serum and supplements for starvation. Briefly, the Matrigel stored at −20 °C was thawed in ice to prevent premature polymerization. Aliquots of 50 µl were plated into each well of pre-chilled 96-well culture plates and were left to polymerize at 37 °C for 2 h. HUVECs were removed from confluent cultures by treatment with Trypsin 0.05% (HyClone, Logan, UT, USA). HUVECs were collected using serum-containing media to stop trypsin effect, then counted, centrifuged and washed with phosphate buffer saline, then resuspended with CdM obtained from the different hypoxic episodes, including intermittent shots (INTR.10, INTR.20, INTR.30, INTR.40) and continuous shots (CONT.5, CONT.10, CONT.15) of hypoxia and from its normoxic counterparts as a control in triplicates. HUVECs were seeded at a density of 2 × 104 cells/well. For quantification of tube formation complex, three primary variables were used to determine the magnitude of tube formation, average of total length of the branched tube, the number of loops and covered area. These variables were measured by Wimasis Image Analysis software.

VEGF quantification by enzyme-linked immunosorbent assay

The CdM that was collected as described above, was used for the determination of VEGF levels secreted by CSCHY P and CSCNOR using the Enzyme-Linked Immunosorbent kit. Human VEGF quantitative ELISA assay (R&D Systems, Minneapolis, MN, USA) was performed according to manufacturer’s instructions. A quantity of 200ul of the CdM from different CSCsHY P treatments was tested.

Cell migration assay of HUVECs

HUVECs ability to migrate under the treatment of the CdM was investigated using a Transwell plate of 8 um pore-size chamber (Corning, Corning, NY, USA). HUVECs were starved overnight using EBM-2 media serum and supplement free for 6 h, then seeded in the upper chamber (insert) of the 24 well plate at density of 5 × 104 cells in 500 uL of EBM -2 media, each placed into lower wells containing 750 ul of CdM obtained from (INTR.10, INTR.20, INTR.30, INTR.40) intermittent hypoxic shots in addition to continuous hypoxic shots (CONT.5, CONT.10, CONT.15). The same procedure was applied to the control by using serum-free EBM-2 media in the lower well instead of CdM. Then, cells were allowed to migrate for 8 h, then 70% cold ethanol was used for 10 min to fix the chambers. Membranes were stained with 0.5% (w/v) crystal violet (Sigma-Aldrich, St. Louis, MO, USA) for 30 min and afterward washed carefully with water. Cotton swabs were used to remove the cells that were not able to migrate to lower wells. Solubilizing bound crystal violet in methanol 100% (wt./vol) was used to quantify the migrated cells. Cells were inspected under an inverted light microscope (Olympus, Tokyo, Japan) and photographed at 4×  magnification. Each sample was done in triplicate and three fields were measured from each to calculate the migration.

Direct and indirect co-culture of HUVECs and CONT.5 CSCs mammospheres

A co-culture system has been used to better understand how CSCs HY P regulate their microenvironment and interact with HUVECs as a vascularization model for angiogenesis and migration.

Direct co-culture

HUVECs were seeded on 0.1% gelatin (Sigma, St. Louis, MO, USA) coated 6-well plate until 40% confluency in EGM-2 media. From the CSCsHY P subpopulation exposed to CONT.5, a single hypoxic CSC was co-cultured in contact with HUVECs. The plates were incubated for few hours to allow CSCsHY P adhere to the wells already seeded with HUVECs. Media was changed to a 1:1 mixture of EBM-2 and DMEM/F-12 free from serum and supplements and incubated for 72 h.

Indirect co-culture

Transwell plates of 0.4 µm pore size chambers were seeded with HUVECs in lower wells after coating them with 0.1% gelatin, while CSCsHY P that were exposed to CONT.5 were seeded in the upper chamber as co-cultured in a 1:1 mixture of EBM-2 and DMEM/F-12 free from serum and supplements.

Control HUVECs were cultured in EBM-2 without serum and without supplements in both the direct and indirect experiments.

Wound healing assay for HUVECs

In a 6-well plate, HUVECs at a density of 12 × 104 were seeded till confluency. Then they were incubated for 6 h in EBM-2 media without serum to prevent any external factors from contributing to wound closure. After 6 h, using a 200uL micropipette tip, a straight wound-like scratch was made into the cells. Then cells were treated with CSCsHY P CdM obtained from INTR.10, INTR.20, INTR.30, INTR.40, CONT.5, CONT.10 and CONT.15 hypoxic shots in which CSCsHY P were grown, in addition to the control which was treated with CdM obtained from CSCsNOR. Cells were carefully washed using phosphate buffer saline (HyClone, Logan, UT, USA), before and after making the scratch. Wound healing was inspected under an inverted light microscope (Olympus, Tokyo, Japan) and photographed after 48 h and at the beginning of the experiment. The surface area of the wound could not be calculated by image analysis software because control cells started to die after 16 h.

Wound healing assay for Hypoxic CSCs

In a 6-well plate coated with 1% gelatin, CSCs were seeded at a density of 8 × 104 and incubated overnight, using DMEM/F-12 media without serum to prevent any external factors from contributing to wound closure. The subpopulation of CSCs used was originally cultured under intermittent or continuous hypoxic conditions; INTR.10, INTR.20, INTR.30, INTR.40, CONT.5, CONT.10, and CONT.15 in addition to the control CSCsNOR originally grown under normoxia. They were all used upon confluency before being seeded in a gelatin-coated 6-well plate to allow CSCs adherence as a 2D monolayer for wound healing assay. Using a 200 uL micropipette tip, a straight wound-like scratch was made. Cells were washed using phosphate buffer saline, before and after making the scratch, and then relevant media was used; DMEM/F-12 without serum. Wound healing was photographed at the beginning of the experiment and after 36 h. The surface area of the wound was measured by image analysis (ImageJ 1.4.3.67 Launcher Symmetry Software).

Statistical analysis

Results were presented as a mean ± SD. Experiments were performed in triplicate with comparable results unless indicated otherwise. An ANOVA test was used to compare the differences between replicates which were considered significant at p ≤ 0.05. GraphPad Prism (version 7.04; San Diego, CA, USA) was used for statistical analysis.

Results

(3D) Mammospheres Derived from MCF-7 cells

Mammosphere culture has been widely used to enrich CSCs. Thereof, MCF-7 mammospheres were cultured in ultra-low adherence plates and flasks, in serum-free media with supplements and successfully formed rigid and compact (3D) structures within 3–7 days (Fig. 2). The mammosphere multicellular spheroidal clumps became larger in size at day 7 compared to earlier days of growth as shown in (Fig. 2) with 30 um scale bar comparison. Moreover, as a quantitative measure, at day seven the mammosphere formation efficiency (MFE) of MCF-7 cells was 0.012% (±0.002%) only and after three weeks the MFE had increased to 5.30% (±0.307%).

Figure 2 The morphology of MCF-7 mammospheres.

(A) MCF-7 floating cells in serum-free low adherent conditions required for mammospheres formation. (B) At day 3 the (3D) mammospheres began to structure. (C) At day 7 larger and denser 3D mammospheres have structured in a rigid and compact form. Representative images magnification (10×  objective) and 30 um scale bar by Olympus inverted microscope.

Sorting of CSCs From 3D mammospheres using magnetic beads

The magnetic beads separation protocol uses antibodies conjugated to a magnetic bead. CSCs were successfully sorted and isolated using the magnetic beads technique, but their number became sufficient only 3 days after sorting. Thereafter, sorted CSCs were identified via FACS analysis and were used for hypoxic-condition exposure. The expression of CD44+/CD24− mammospheres is shown in (Table 1) and their morphology is shown in (Fig. 2).

Table 1 The summary of CD44+/CD24− expression obtained by flow cytometry among different MCF-7 sub-population derived cells.

Cells subpopulation	CD44+/CD24−expression %	
Parent MCF-7 Cells	1.0 (±0.1%)	
Unsorted Mammospheres (day 7 in ultra-low attachment, serum free media)	33.2 (±3.0%)	
Sorted CSCs mammospheres (3 days after sorting)	81.0 (±7.5%)	
Sorted CSCs mammospheres (After a total of 21 days)	35.5 (±4.5%)	
CSCsHY P exposed to INTR.20 shots	39.8 (±5.2%)	
CSCsHY P exposed to CONT.5 shots	51.6 (±6.1%)	
CSCsHY P exposed to INTR.40 shots	0.3 (±0.01%)	
CSCsHY P exposed to CONT.15 shots	0.5 (±0.05%)	

Enrichment of sorted CSCs mammospheres under Hypoxic/Normoxic conditions

The morphology of CSCsHY P and the control in normoxia CSCsNOR is shown in (Fig. 3). It is observed that CSCsHY P that underwent CONT.5 and INTR.20 hypoxic shots were denser and larger in number than CSCsNOR. CSCsHY P that underwent CONT.15 and INTR.40, that is the maximum and final hypoxic shots showed the least growth compared to other hypoxic exposures and CSCsNOR. The MFE could not be measured for CSCsHY P because long-term hypoxic exposure (INTR.40 and CONT.15) made mammospheres aggregate into clumps, thus, the single mammosphere could not be distinguished and compared to control and to other hypoxic mammospheres.

Figure 3 The 3D morphology of CSCs mammospheres under hypoxic condition (B, C, D, E) or in normoxia as in (A).

The (3D) morphology of CSCs clustered in multicellular spheroids in serum-free media. (A) CSCs under normoxia. (B) and (C) CSCs exposed to INTR.20 and INTR.40 hypoxic shots respectively. (D) and (E) CSCs exposed to CONT.5 and CONT.15 hypoxic shots respectively. CSCsHY P that underwent CONT.5 and INTR.20 were denser and larger in number than CSCsNOR. CSCsHY P that underwent CONT.15 and INTR.40 had the least growth compared to other hypoxic exposures and CSCsNOR. Representative images magnification (10×  objective) and 30 um scale bar by Olympus inverted microscope.

Identification of CD44+/CD24− phenotype content by flow cytometry

Flow cytometry conducted for CSCs after 3 days of sorting showed a percentage of CD44+/CD24− expression equal to 81.0 (±7.5%). Three weeks (21 days) after growing in ultra-low attachment flasks, the percentage decreased to approximately 35.5 (±4.5%). In contrast, the parental MCF-7 cells had only 1 (±0.1%) expression and the unsorted mammospheres had 33.2 (±3%) expression. After exposure to hypoxic conditions, flow cytometry results based on CD44+/CD24− markers showed that the highest expression occurred in CSCsHY P exposed to INTR.20 shots, 39.8 (±5.2%), and in CSCsHY P exposed to CONT.5, 51.6 (± 6.1%). On the other hand, the lowest percentages were observed in CSCsHY P that underwent or were exposed to INTR.40 shots, 0.3 (±0.01%), and in CSCsHY P exposed to CONT.15, 0.5 (±0.05%). In conclusion, isolated and sorted CD44+/CD24− breast cancer cells only temporarily preserve this phenotype and ultimately revert to an equilibrium state in which the expanded sub-population regains the original cell surface profile of the parental cell line, which was deduced because the unsorted mammospheres and the 3-week old sorted CSCs both have similar CD44+/CD24− content (33.2% and 35.5%, respectively). The CD44+/CD24− percentage in unsorted mammospheres that were grown in ultra-low attachment plates in serum-free media for 7 days before sorting was 33.2 (±3%), which is much higher than the percentage of parental MCF-7 cells 1.0 (±0.1%). The expression of CD44+/CD24−content for all subpopulations is summarized in (Table 1) and demonstrated in (Fig. 4).

Figure 4 Determination of the cancer stem cell-like properties in mammosphere forming cells based on surface marker CD44+/CD24−.

(A) parent MCF-7 cells. (B) unsorted mammospheres. (C) sorted CSCs mammospheres after 3 days of sorting. (D) CSCs mammospheres after 21 days of sorting. (E) CSCsHY P underwent INTR.20 shots. (F) CSCsHY P underwent CONT.5 shots. (G) CSCsHY P underwent INTR.40 shots. (H) CSCsHY P underwent CONT.15 shots.

Hypoxic CSCs show higher drug resistance to conventional chemotherapies

To examine whether self-renewing CSCs mammospheres have higher chemoresistance ability, and to assess if hypoxic treatment of CSCs would increase their stemness character and chemoresistance, MTT assay was implemented to test the sensitivity of parental MCF-7 versus the sorted CSCs mammosphere towards doxorubicin. In addition, we assessed the resistance towards doxorubicin acquired by CSCsHY P which underwent intermittent hypoxic shots (INTR.10, INTR.20, INTR.30 INTR.40), and CSCsHY P that underwent continuous hypoxic shots (CONT.5, CONT.10, CONT.15). Then we compared the inhibitory drug concentration (IC50) results for the CSCsHY P versus CSCsNOR mammospheres (as control) and compared the IC50 value of CSCsHY P versus parental MCF-7 cells, described in Table 2. Overall, the drug resistance of CSCsNOR was found to be higher (IC50 = 1.90 uM) when compared to the parental MCF-7 cells (IC50 = 0.46 uM) by 4.08 times. Comparing the IC50 values of CSCsNOR mammospheres with CSCsHY P, we observed significantly increased values under hypoxic conditions at INTR.20 (IC50 = 6.13 uM) and CONT.5 (IC50 = 7.12 uM) with (3.28) and (3.76) fold increases respectively compared to IC50 of CSCsNOR mammospheres. These increaseswere much higher upon comparing IC50 for CSCsHY P with parental MCF-7. The lowest IC50 values were observed at INTR.40 and CONT.15 which represent the final hypoxic treatments. Taken together, these results support the likelihood that enriching the mammospheres’ culture with CSCs contributed to the higher drug resistance toward doxorubicin when compared to the parental MCF-7.

Table 2 The IC50 values (uM) of doxorubicin against MCF-7 parental, CSCsNOR mammospheres and CSCsHY P of different hypoxic shots after 72 h treatment.

Cell subpopulation	IC50 (uM)± STD	Fold change comparedto sorted normoxic mammosphere (control)	Fold change compared to Parental MCF-7	
MCF-7 parental	0.46 (±0.06)	N/Ab	N/Aa	
CSCsNOR mammospheres (control)	1.90 (±0.35)	N/Aa	4.08 ↑	
CSCsHY P exposed to INTR.10	4.64 (±0.15)	2.45 ↑	10.00 ↑	
CSCsHY P exposed to INTR.20	6.2 (±0.29)	3.28 ↑	13.39 ↑	
CSCsHY P exposed to INTR.30	4.101 (±0.22)	2.16 ↑	8.84 ↑	
CSCsHY P exposed to INTR.40	0.334 (±0.02)	0.18 ↓	0.72 ↓	
CSCsHY P exposed to CONT.5	7.124 (±0.19)	3.76 ↑	15.35 ↑	
CSCsHY P exposed to CONT.10	4.486 (±0.11)	2.36 ↑	9.67 ↑	
CSCsHY P exposed to CONT.15	0.441 (±0.04)	0.23 ↓	0.95 ↓	
Notes.

All data are expressed by means ± SD; P < 0.0001 compared to the parental group.

a N/A: Value not calculated; not applicable to compare the reference to its value.

b N/A: Value not calculated because we refer and compare the IC50 of sorted CSCsNOR mammospheres to MCF-7 Parental cells IC50 not the contrary.

N/A not applicable

STD standard deviation

uM microMolar

↑ increased

↓ decreased

Hypoxic conditioned medium (CdM) induces HUVECs to form capillary-like tube structures in matrigel

In this study, HUVECs were treated with CdM in which CSCs were expanded under several hypoxic and normoxic conditions, as described in the method section. Our results showed that treating HUVECs with CdM obtained from CSCsHY P resulted in an increased construction of capillary-like tube structures that reached its peak at 8 h of treatment and disappeared eventually in a time-dependent manner. However, HUVECs treated with CdM obtained from CSCsNOR (control), resulted in a reduced formation of complex tubular structures compared to the ones treated with the CdM obtained from CSCsHY P (Fig. 5). The capillary-like tube structures were evaluated by quantitative analysis using Wimasis Image Analysis. The results showed that the highest increase of branched tube lengths, the number of loops and the covered area was with HUVECs that were cultured with CdM obtained from CSCsHY P culture that underwent INTR.20 and CONT.5 shots at each time point (as demonstrated in Figs. 5 and 6). A bar graph (data combined from three independent trials) (Fig. 6) shows that “the total length of the tubes” underwent a remarkable increase upon the treatment with CdM from INTR.20 and CONT.5, by almost 2.4 fold in both INTR.20 and CONT.5 when compared to the control, moreover, the “covered area” significantly increased when HUVECs were cultured with CdM from CSCsHY P culture that underwent CONT.5 and INTR.20, by 3.5 and 2.7 times respectively. The total number of loops were significantly increased by 6.7 fold in INTR.20 and 8.9 fold in CONT.5 compared to control HUVECs incubated with CdM obtained from the CSCsNORculture expanded in normoxic state. As shown in Figs. 5 and 6, CdM induced a significant increase in tube formation in HUVECs starting from the treatment with INTR.10 CdM and reaching a maximum at INTR.20 (Fig. 5F), whereas CONT.5 (Fig. 5B) caused the highest increase in tube formation.

Figure 5 Effect of normoxic and hypoxic CdM on capillary-like tube structure formation in HUVECs.

HUVECs were cultured in CdM obtained from CSCsHY P mammospheres that were previously grown and exposed to (B) CONT.5 (C) CONT.10 (D) CONT.15 (E) INTR.10 (F) INTR.20 (G) INTR.30 (H) INTR.40 hypoxic shots and compared to HUVECs cultured using CdM obtained from CSCsNOR as a control as shown in (A). Capillary-like tubular structures were the highest in using CdM obtained from CONT.5 and INTR.20 exposure. Representative images magnification (4×  objective) Olympus inverted microscope.

Figure 6 Assessment of capillary-like tubular structure formation.

(A) Total tube lengths formed. (B) Number of loops intersections. (C) Covered area percentage. The results represent mean ± SD of three independent experiments. The symbol marks the statistical significance levels as follows: (*) indicates p < (0.001) and (∗∗) indicates p < (0.0001) as compared to the control.

Detection of secreted vegf in conditioned-medium (CdM) by ELISA

In order to identify the possible mediators of angiogenesis in breast cancer, a quantitative assay was performed for the main angiogenic factor, VEGF-A. As the CdM obtained from the hypoxic episodes INTR.20, INTR.30 and CONT.5 had the highest VEGF concentrations, INTR.20, 1159 (±90 pg/ml), INTR.30, 1013 (±32 pg/ml) and CONT.5, 934 (±28 pg/ml) are compared to control normoxic CdM 225 (±24 pg/ml) as shown in (Fig. 7).

Figure 7 VEGF-A concentrations secreted by CSCsHY P or CSCsNOR were determined by ELISA assay.

The results represent mean ± SD of three independent experiments. Statistically significant differences among the treated versus control cells are shown by (*) (p < 0.05) and (∗∗) (p < 0.0001).

Figure 8 Transmigration of HUVECs induction by CdM obtained from CSCsHY P compared to control where EBM-2 serum-free served as control.

CdM from hypoxic shots CONT.5 and INTR.20 proved to be a potent chemoattractant compared to the control. All values were determined as averages of three independent experiments. The results represent mean ± SD of three independent experiments. Statistically significant differences among the treated and control cells are shown by (∗∗) (p < 0.0001).

Effect of conditioned-medium (CdM) on migration of HUVECs

The role of the CdM in promoting migration of HUVECs was investigated using migration assay to confirm whether secretions of CSCsHY P enhance angiogenesis in the tumor microenvironment. HUVECs were tested for their ability to migrate toward CdM from different hypoxic episodes compared to their migration toward endothelial basal medium (EBM-2) serum and supplement free, as a control, using transwell migration assay. The Transwell migration assay is used to create a chemical gradient by putting the CdM in the lower chamber. Comparing the migration of HUVECs from upper chamber across the membrane upon using the EBM-2 without serum and without supplements (control) group versus using the CdM obtained from hypoxic shots, it was found that migration was much higher under the use of the hypoxic CdM (Fig. 8). The assay results showed that there were more purple-stained cells migrating toward the CdM obtained from the CONT.5 shot than the other two episodes (CONT.10 and CONT.15). The CdM obtained from the INTR.20 episode had also more purple-stained cells migrating towards it, compared to the other three episodes (INTR.10, INTR.30, and INTR.40). Interestingly, a significant difference found was that the migrated-cell count in HUVECs towards CdM from CONT.5 vs control was 45,000 (±2.1) vs. 4,500 (±1.5), while the migrated cell count in INTR.20 vs control was 38,000 (±2.6) vs. 4,500 (±1.5). These findings suggest that the secretions presented in the CdM which were obtained from CONT.5 and INTR.20 episodes significantly promoted the migration of HUVECs. The data also showed that the migration capacity of HUVECs increased significantly (10-fold) when cultured using the CdM from CONT.5 shot and (8.4-fold) when cultured using the CdM from INTR.20 shot compared with HUVECs cultured with control (Fig. 8).

Stimulation of HUVECs growth & morphological alterations in co-culture with CSCsHY P

To clarify the interaction between endothelial cells and CSCsHY P in the tumor microenvironment, HUVECs were examined in direct and indirect co-cultures with the CSCsHY P subpopulation that underwent CONT.5 episode. The control of HUVECs was cultured in serum and supplement free media and possessed a ‘teardrop-like’ morphology. In the direct co-culture model, the endothelial cell morphology changed and acquired elongated threadlike shape as a response to the influence of CSCsHY P secretions though the growth media was free from serum and any growth factors as shown in (Figs. 9A–9C).

Figure 9 Morphological changes of HUVECs after co-cultured with CSCsHY P subpopulation that underwent CONT.5.

Hypoxic episode (A–C) for direct co-culture and (D–E) for indirect co-culture. Direct co-culture model: (A) CSCsHY P mammospheres were seeded directly above HUVECs monolayer at (0) time. (B) After 36 h HUVECs proliferate normally with CSCsHY P. (C) at 72 h HUVECs obtained elongated threadlike shape mediated by CSCsHY P secretions. Indirect co-culture where permeable membrane was placed between HUVECs and CONT.5 CSCsHY P subpopulation: (D) HUVECs after 36 h. (E) At 72 h HUVECs morphology changed to form net-like structures resembling a vascular network as mediated by CSCsHY P secretions that passed through the membrane. Representative images magnification (4×  objective) Olympus inverted microscope.

We also examined the growth of HUVECs indirectly co-cultured with CSCsHY P using a 0.4 uM permeable membrane that was placed between HUVECs and CSCsHY P subpopulation. Indirect co-culture of the subpopulation of CONT.5 CSCsHY P with HUVECs induced sequential morphological changes. As a response to the influence of secretions from CSCsHY P thattrans-passed through the membrane, the morphology of HUVECs began to change at 72 h, forming net-like structures resembling a vascular network, though the growth media was free from serum and any growth factors (Figs. 9D & 9E). Control HUVECs proliferation reached a plateau in 24 h after which cells started to die.

In vitro wound-healing assay of HUVECs

This assay was done to assess how secretions from CSCsHY P in the tumor microenvironment would affect the ability of HUVECs to migrate in response to chemoattractants in the CdM obtained from different hypoxic shots. This model mimics the tumor microenvironment where angiogenic ability of endothelial cells is affected by secretions from CSCsHY P. In the control experiment, HUVECs were treated with CdM obtained from CSCsNOR. In the presence of the CdM obtained from hypoxic shots, HUVECs were able to migrate and the wound closure was enhanced after 48 h. The wound closure was significantly enhanced by CdM obtained from INTR.20 (Fig. 10B) and CONT.5 (Fig. 10C) shots. Whereas in the case of the control, HUVECs that were treated with CdM media obtained from CSCsNOR, showed a weaker capacity to migrate. The wound surface area could not be calculated in order to assess wound closure because the HUVECs treated with CdM obtained from CSCsNOR and from CSCsHY P after long-term exposure (INTR.40 and CONT.15) started to die after 16 h of treatment which prevented further quantitative measurements.

Figure 10 HUVECs cell migration assessment using wound-healing assay.

The upper group of images represents the wound at zero time of treatment with CdM while lower images represent the wound after 48 h of treatment with CdM. (A) control treated with CdM obtained from CSCsNOR. (B) Treated with CdM from INTR.20 shot. (C) Treated with CdM from CONT.5 shot. Representative images magnification (4×  objective) Olympus inverted microscope.

In vitro wound-healing assay of hypoxic CSCs

Due to the significant role that CSCs play in tumor recurrence and metastasis, a wound-healing assay was performed. CSCs were seeded as a (2D) monolayer, scratched, and then cell migration was evaluated. The seeded CSCs were originally expanded and grown in hypoxic conditions, INTR.10, INTR.20, INTR.30, INTR.40, CONT.5, CONT.10, and CONT.15 in addition to the control CSCs in normoxia. The scratched CSCsHY P and CSCsNOR cultured in DMEM/F-12 without serum were assessed for wound closure ability which was determined by calculating the ratio of the wound surface area at the endpoint (36 h) to its surface area at the starting point of the experiment (zero hour). Images were analyzed by ImageJ software. Subpopulations of CSCsHY P showed greater ability for wound closure compared to CSCsNOR (Figs. 11 and 12). CSCsHY P from INTR.20 and from CONT.5 manifested the greatest wound closure (Fig. 11). At 36 h, the wound closure of the control was only 40%, while it was 98% and 95% for INTR.20 (Fig. 12C) and CONT.5 (Fig. 12F), respectively.

Figure 11 Wound healing of CSCs.

The upper group of images demonstrate the wound surface area of CSCs monolayer at the time of wound scratching (zero time). The lower group of images are for wound closure after 36 h from scratching. (A) CSCsNOR used as control. (B–H) CSCsHY P exposed to various hypoxic treatments. CSCsHY P that were exposed to INTR.20 shown in (C) and CONT.5 shown in (F) both have had the highest wound closure capacity compared to the control CSCsNOR in (A). Representative images magnification (4×  objective) Olympus inverted microscope.

Figure 12 CSCs wound closure quantification.

The quantification of wound closure was measured by ImageJ software and represented in bars expressing the percentage of gap closure. CSCs subpopulation that was exposed to INTR.20 or CONT.5 manifested the highest wound closure capacity compared to the control CSCsNOR. After 36 h, the wound closure was 98% for INTR.20 and 95% for CONT.5 whereas 40% for control. The results represent mean ± SD of three independent experiments. Statistically significant differences among the treated and control cells are shown by (*) (p < 0.001) and (**) (p < 0.0001).

Discussion

The initial stage of our study focused on the formation and characterization of MCF-7 mammospheres enriched with CSCs. The mammosphere culture is an approach to culturing cancer cells in a 3D conformation in vitro. These 3D models have been well recognized in breast cancer research due to the great similarity of these models to the in vivo solid tumors (Weiswald et al., 2015; Boo et al., 2016). The 3D models are not only biologically spherical in shape, but also possess many shared features with solid tumors which were not observed in traditional 2D monolayer cultures (Baker & Chen, 2012; Ekert et al., 2014). Interestingly, several current reports have shown that CSCs are particularly enriched and maintained in cultures of 3D mammospheres (López et al., 2012; Liu et al., 2013). CSCs show a tendency to differentiate in the availability of serum, accordingly serum-free culture conditions are believed to maintain the CSCs in an undifferentiated stage supporting their enrichment, with recognizable loss and reduction in the size of the upper cell in the hierarchical structure (Kruyt & Schuringa, 2010; Lin et al., 2012). The results of our study have shown a CD44+/CD24− expression (Table 1) for parental MCF-7 of (1%). After being cultured in a low-adherence, serum-free, with supplements media, CSCs were sorted and again identified with (FACS) and showed an extreme elevation of CD44+/CD24− expression to (81.3%) only three days after being grown in mammosphere culture conditions, while it was (33.2%) before being sorted as CSCs in mammosphere culture media. Interestingly, it reverted to (35.5%) at day 21. From these results, it is concluded that sorting of CSCs allows them to acquire stemness character that is much higher than their parental cells and their unsorted population, and declines over time. The interpretation is that CSCs upon repetitive and continuous growth are driven towards differentiation into regular cancer cells instead of bearing a CSC state. In another interpretation, a distinctive characteristic of mammosphere culture conditions -unlike the adherent cell culture is that during the passaging of CSCs most of them die early (through apoptosis/anoikis) and the mitogen-responsive anoikis resistant cells that were supposed to be CSCs grow and form new mammospheres. Therefore, increased expression of CD44+/CD24−within mammospheres compared to parent cells could be due to an enriched anoikis-resistant proliferation process (Deleyrolle et al., 2011).

The second stage of our study was to investigate the role of cycling hypoxia on CSCs propagation, stemness character, drug resistance, and adequate motility to recruit new tumor growth at distant sites through migration and induction of VEGF expression level. In our study as detailed in methods, we divided CSCs into two groups to be exposed to intermittent or continuous shots of hypoxia. MTT cytotoxicity assay was performed to assess chemoresistance of CSCs in hypoxia vs. normoxia vs. parent MCF-7 cells using doxorubicin. Doxorubicin is a cytotoxic anthracycline, that is widely used drug to treat breast cancer. The cytotoxic effect of doxorubicin on malignant cells is due to nucleotide base intercalation and cell membrane lipid binding. Intercalation inhibits nucleotide replication and action of DNA and RNA polymerases. The interaction of doxorubicin with topoisomerase II to form DNA-cleavable complexes appears to be an important mechanism of doxorubicin cytocidal activity (Miglietta et al., 2000). Our results have shown that parent MCF-7 had the lowest drug resistance to doxorubicin (IC50 = 0.46 uM) compared to normoxic CSCs mammosphere (IC50 = 1.90 uM), and when both were compared to CSCsHY P that underwent hypoxic episodes of INTR.20 and CONT.5, they had (IC50 = 6.20 uM) and (IC50 = 7.12 uM), respectively. Consequently, one concludes that hypoxia remarkably elevates chemoresistance not only upon comparison with parent MCF-7, but also upon comparison with the CSCsNOR (control). It is noteworthy to explain that the IC50 values of CSCsHY P subjected to all these hypoxic episodes (INTR.10, INTR.20, INTR.30, CONT.5, and CONT.10) were higher than parental MCF-7 and than CSCsNOR (control), with an exception for the CSCsHY P that underwent (INTR.40 and CONT.15) episodes representing the finale of the hypoxic treatment which had IC50 values that were less than the IC50 of CSCsNOR (control) and close to parental MCF-7. It is notable to compare MTT assay results with flow cytometry results of the CSCsHY P as they reflect one another. MTT assay results for sorted CSCs have shown higher chemoresistance with (4-fold) increased IC50 compared to parental MCF-7 whose CD44+/CD24− expression was (1%). This proposes the likelihood that increased stemness is the contributor to the higher drug resistance in mammospheres as suggested previously (Calcagno et al., 2010). Moreover, identification of CD44+/CD24− expression by flow cytometry applied on CSCsHY P that underwent CONT.5 and INTR.20 revealed their possession of the highest expression at 51.6% and 39.8%, respectively. Among all hypoxic episodes, the lowest expressions of 0.5% and 0.3% were for CSCsHY P that underwent CONT.15 and INTR.40, respectively. Interestingly, CSCsHY P that underwent CONT.5 and INTR.20, which had the highest CD44+/CD24− expression also had the highest chemoresistance results in contrast to CONT.15 and INTR.40 which had the lowest CD44+/CD24− expression, and also the lowest chemoresistance results. These results agree with their counterparts in past studies which showed that chemoresistance is closely related to many fundamental or acquired characteristics of CSCs, such as DNA repair ability, quiescence, overexpression of antiapoptotic enzymes, drug efflux transporters and detoxifying enzymes (Vinogradov & Wei, 2012). Our findings also agree with the study of Hermann et al. (2007), who reported the existence of a subpopulation of pancreatic cancer cells which resembled stem cells and had higher gemcitabine resistance both in vitro and in vivo. Since CSCs play a significant role in tumor recurrence and metastasis, wound-healing was investigated to evaluate the behavior of CSCs after being exposed to different hypoxic shots. The adherent CSCsHY P showed greater ability of self-renewal and wound healing, especially after being exposed to INTR.20 and CONT.5 hypoxic shots. They also showed the slowestwound healing rate after being exposed to INTR.40 and CONT.15. The wound healing results were mirror-image reflection of the chemoresistance and stemness characteristics results relevant to each hypoxic episode discussed above.

The third phase of this study was performed to evaluate the role of secretions of CSCs after being subjected to different hypoxic shots on angiogenesis using (HUVECs) as a model proposed to modulate neovascularization phenotype. Increased VEGF secretion in CSCsHY P versus control CSCsNOR was confirmed. This finding demonstrated that CSCsHY P secretions increased the level of VEGF significantly which is linked to the angiogenic potential of the CdM reported in this study. In this manner, the study of Bao et al. (2006) showed that CD133+ enriched stem cell-like glioma cells (SCLGC) produced tumors with an increased tumor vascularity, necrosis, and hemorrhage. Also, the authors showed that the VEGF expression was 10-20 fold up-regulated in CD133+ SCLGC.

The direct and indirect co-culture of CSCsHY P that underwent CONT.5 with endothelial HUVECs proved that this co-culture system comprised constituents that promoted angiogenic phenotype, such as proliferation and differentiation of endothelial cell into a net-like structure. Interestingly, the influence of the secretions from CSCsHY P modified the HUVECs morphology to a longer, mesenchymal-like appearance and enhanced their migration rate. The migration assay confirmed that CdM obtained from CSCsHY P enhanced the migration rate of HUVECs compared to its control group cultured in EBM-2. HUVECs migrated towards the chemoattractants that CSCsHY P secreted in the CdM that were obtained from all hypoxic episodes, except the CdM that were obtained from CONT.15 and INTR.40, which did not enhance HUVECs migration. However, migration was distinctive upon the use of CdM obtained from CONT.5 and INTR.20. These results could be interpreted in light of the biological process in which the growth of a new blood vessel from pre-existing blood cells via “sprouting” of endothelial cells is led by a tip cell which is a single endothelial one. Tip cells drive vascular growth by detecting gradient proangiogenic mediators such as VEGF. Adjacent endothelial cells become stalk cells that can proliferate and migrate towards the tip cell which results in the elongation of the sprouting vessel (Gerhardt et al., 2003).

The wound healing assay has confirmed that the CdM obtained from CSCsHY P enhanced the repair of the scratch in HUVECs, and closure was significantly higher upon the use of CdM obtained from CONT.5 and INTR.20 episodes in comparison to wound closure treated with CdM media obtained from CSCsNOR. Moreover, the addition of CdM obtained from CSCsHY P cultures into HUVECs was also found to promote the endothelial cell proliferation and induce capillary-like tube structure formation by increasing total length of the formed branched-tube, the number of loops and the area covered by the tubes. This observation was distinctive in the use of CdM obtained from CSCsHY P exposed to CONT.5 and to INTR.20. Together, these results suggest that CSCsHY P cells produce pro-angiogenic factors that may directly alter the behavior of endothelial cells. Such findings are in agreement with the study of (Tu et al., 2009) which showed that A549 lung cancer cells hypoxic CdM improved HUVECs’ cell presence and wound healing migration ability.

In conclusion, the angiogenic components and the angiogenic switch were enhanced by the tumor microenvironment constructed by the CdM of CSCsHY P particularly after five hypoxic episodes of continuous 72-hours weekly exposure (CONT.5) and 20 episodes of intermittent 8-hours exposure to hypoxia, 3 times weekly (INTR.20).

In the context of the above-detailed findings and upon comparing the results of CSCs exposed to intermittent versus continuous hypoxic condition in terms of stemness, chemoresistance, tube formation and VEGF protein expression related to angiogenesis, wound healing and finally, cell migration, the results implied similarity among intermittent and continuous treatments between the early intervals (INTR.20 and CONT.5), between the intermediate intervals (INTR.30 and CONT.10), and also between the late intervals (INTR.40 and CONT.15).

The present study agrees with the study of (Liu et al., 2017) which demonstrated that intermittent and continuous hypoxic conditions significantly increased the migration of MDA-MB-231 breast cancer cells to a certain extent, either because of excessive long-term exposure or because of the increased number of hypoxia-reoxygenation cycles. In this study, the maximum increase of these features by hypoxic effect was observed at early intervals (INTR.20 and CONT.5), whereas it started to decline at intermediate ones (INTR.30 and CONT.10), and drastically declined at late intervals (INTR.40 and CONT.15) to a rate even lower than the control group of CSCs in normoxia

The previous study demonstrated that excessive long-term or high-frequency exposure to hypoxia leads to the generation of reactive oxygen species (ROS), which in turn, induces lipid peroxidation and increases the production of stress responding proteins, thus promotes DNA strand breakage, cellular injury, and apoptosis (Pires et al., 2010; Zepeda et al., 2013). Generally, low levels of ROS are required for stem cells to maintain quiescence and self-renewal. Therefore, in the CSCs scenario, a low concentration of ROS enhances their stemness and contributes to tumorigenesis (Liu & Wang, 2015). However, increased ROS production due to extensive hypoxia causes CSCs exhaustion, differentiation, senescence, and apoptosis (Zhou, Shao & Spitz, 2014).

Thus, it is interpreted that the ROS level began to increase starting from the intermediate intervals (INTR.30 and CONT.10) and reached maximal rate in the late intervals (INTR.40 and CONT.15) proposing the manifestation of differentiation, senescence, and apoptosis in CSCs extensively exposed to hypoxia, based on the previous explanation.

Conclusion

This study has proven that the highly optimized hypoxia-reoxygenation system applied to CSCs has enriched their growth, enhanced their properties of self-renewal, increased the expression of stemness surface markers, and increased their chemoresistance. The findings of this study also demonstrated that the CSCsHY P microenvironment plays a major role in enhancing angiogenesis and tumor vascularization by recruiting endothelial cells into the tumor microenvironment, and in activating distinct molecular mechanisms that need further investigations. Whether the elevation of stemness in CSCsHY P results from the de-differentiation of differentiated MCF-7 cells, or from the enhanced proliferation of CSCs after hypoxic exposure is still unclear and needs further study. The interactions between CSCs in tumor microenvironment and other contributing processes such as hypoxia and angiogenesis are still at the infancy stage and need in-depth future research.

Supplemental Information

Supplemental Information 1 Identification of CD44+/CD24- phenotype content by flow cytometry

The raw data of identification CD44+/CD24- phenotype content by flow cytometry.

Click here for additional data file.

Supplemental Information 2 The IC 50 values (uM) raw data

The raw data of IC 50 values (uM) of doxorubicin against MCF-7 parental, normoxic CSCs mammospheres and hypoxic CSCs mammosphere of different hypoxic shots after 72 h treatment.

Click here for additional data file.

Supplemental Information 3 Effect of normoxic and hypoxic CdM on capillary-like tube structure formation in HUVECs (total tube length)

The raw data that represent the effect of normoxic and hypoxic CdM on capillary-like tube structure formation in HUVECs (total tube length).

Click here for additional data file.

Supplemental Information 4 CSCs wound closure quantification

The raw data of wound closure that was measured by ImageJ software and represented in bars expressing percentage of gap closure.

Click here for additional data file.

Supplemental Information 5 VEGF-A concentrations secreted by CSCsHYP or CSCsNOR were determined by ELISA assay

The raw data represent mean ±SD of three independent experiments.

Click here for additional data file.

Supplemental Information 6 The effect of normoxic and hypoxic CdM on capillary-like tube structure formation in HUVECs

The raw data that represent the effect of normoxic and hypoxic CdM on capillary-like tube structure formation in HUVECs (total tube length).

Click here for additional data file.

Supplemental Information 7 Transmigration of HUVECs induction by CdM obtained from CSCsHYP compared to control where EBM-2 serum-free served as control

The raw data of transmigration HUVECs induction by CdM obtained from CSCsHYP compared to control where EBM-2 serum-free served as control.

Click here for additional data file.

Supplemental Information 8 The effect of normoxic and hypoxic CdM on capillary-like tube structure formation in HUVECs

The raw data that represent the effect of normoxic and hypoxic CdM on capillary-like tube structure formation in HUVECs (covered area).

Click here for additional data file.

Supplemental Information 9 Assessment of capillary-like tubular structure formation using Wimasis Image Analysis software

Assessment of capillary-like tubular structure in the control.

Click here for additional data file.

Supplemental Information 10 Assessment of Capillary-likeTubular Structure Formation using Wimasis Image Analysis software

Assessment of Capillary-likeTubular Structure in INTR.10

Click here for additional data file.

Supplemental Information 11 Assessment of capillary-like tubular structure formation using Wimasis Image Analysis software

Assessment of capillary-like tubular structure formation inINTR.20.

Click here for additional data file.

Supplemental Information 12 Assessment of capillary-like tubular structure formation using Wimasis Image Analysis software

Assessment of capillary-like tubular structure formation in INTR.30.

Click here for additional data file.

Supplemental Information 13 Assessment of capillary-like tubular structure formation using Wimasis Image Analysis software

Assessment of capillary-like tubular structure in INTR.40.

Click here for additional data file.

Supplemental Information 14 Assessment of capillary-like tubular structure formation using Wimasis Image Analysis software

Assessment of capillary-like tubular structure formation CONT.5.

Click here for additional data file.

Supplemental Information 15 Assessment of capillary-like tubular structure formation using Wimasis Image Analysis software

Assessment of capillary-like tubular structure formation in CONT.10.

Click here for additional data file.

Supplemental Information 16 Assessment of capillary-like tubular structure formation using Wimasis Image Analysis software

Assessment of capillary-like tubular structure formation in CONT.15.

Click here for additional data file.

Additional Information and Declarations

Competing Interests

Author Contributions

Human Ethics

Data Availability

The authors declare there are no competing interests.

Fuad M. Alhawarat conceived and designed the experiments, performed the experiments, analyzed the data, contributed reagents/materials/analysis tools, prepared figures and/or tables, authored or reviewed drafts of the paper, approved the final draft.

Hana M. Hammad conceived and designed the experiments, analyzed the data, contributed reagents/materials/analysis tools, authored or reviewed drafts of the paper, approved the final draft.

Majd S. Hijjawi conceived and designed the experiments, performed the experiments, analyzed the data, authored or reviewed drafts of the paper, approved the final draft.

Ahmad S. Sharab analyzed the data, contributed reagents/materials/analysis tools, authored or reviewed drafts of the paper, approved the final draft.

Duaa A. Abuarqoub performed the experiments, contributed reagents/materials/analysis tools, authored or reviewed drafts of the paper, approved the final draft.

Mohammad A. Al Shhab contributed reagents/materials/analysis tools, authored or reviewed drafts of the paper, approved the final draft.

Malek A. Zihlif conceived and designed the experiments, analyzed the data, contributed reagents/materials/analysis tools, prepared figures and/or tables, authored or reviewed drafts of the paper, approved the final draft.

The following information was supplied relating to ethical approvals (i.e., approving body and any reference numbers):

Jordan University Hospital granted approval from their Institutional Review Board (IRB), decision number 98/2018 and approval number 67/2018/481.

The following information was supplied regarding data availability:

The raw data is available in Supplemental Files.

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
