# Peer review of "The effect of cycling hypoxia on MCF-7 cancer stem cells and the impact of their microenvironment on angiogenesis using human umbilical vein endothelial cells (HUVECs) as a model"

_PeerJ, doi:10.7717/peerj.5990_

## Round 0.1 · original submission · Major Revisions

Dear Dr Alhawarat,

Please see the reviewer's comments. You will need to address all of the issues that they have raised. This will require additional experiments (for example, neutralizing antibody to VEGF to show that VEGF is responsible for the effects) and a thorough re-writing of the manuscript for clarity and for correcting the language and grammar.

The rationale for the experimental conditions and how these are relevant to human breast cancer needs proper explanations.

Additional verification of the experiments is required, as outlined by the reviewers.

Figures need to be improved.

Yours,

Philip Coates

Reviewer 1 ·

Basic reporting

The work described in the manuscript is clear, nevertheless I would recommend a revision of the English used. Especially, I would suggest to carefully revise the punctuation, as it affects the understanding of the message and also the flow of the reading.
In many cases a ";" is used in the place of "," and several subordinates are not in agremment with the main sentence.

I have found an appropriate descrition of the background with recent literature references.

I would recommend to improve the quality of the figures and of the figure legends.

In my opinion, although the results are ineteresting, the authors need to explain better and elaborate the role of the different treatments the experimentally reproduce, in the light of their finding and give a link of these "arificially induced" hypoxia to the physiological situation that may occur in tissues.

Experimental design

Experiments are clear enoghu and the methods are reported in details.

Validity of the findings

Some conclusions are drawn without a clear and robust evaluation (e.g. the role of VEGF in changing cell shape an migration ability).
I would require a better description of the mammosphere generated, especially giving some quantifiable paramenters, not only a picture to show the appearence of the cells.

Additional comments

The authors make clear which is the purpose of their study at the end of the introduction, following a logical scheme. Nevertheless, they do not give any background about the drug they intend to use (doxorubicin). I suggest to briefly explain the effect of the drug and maybe give a reference.
The workflow shown in Figure 1 is unnecessary, as the experiment flow has been stated at the end of the introduction. To my opinion, the graph does not anything to the description.
Figure 2: it is evident that between day 2 and day 7 the clumps and clusters appear bigger, but is there a positive control to show? Or a specific staining giving evidence that these clumps are actually formed of vital cells? It is also required to label panels with A, B, C etc. as the authors refer to letters in the figure legend. The figure legend should be composed of a title and an explanation that is not the mere copy of the title. Moreover, in the results section (page 16 line 400) the authors state that after three weeks the MEF increased. Can they show a picture to highlight the final material?
Figure 3: I understand it is only a morphological observation, but the authors should give an explanation of the difference they see among the different treatments. There are no parameters or other evaluation of the mammospheres in fig. 3 A-G. What is relevant? How the intermittent shots are different from continuous in term of results?
Table 1: how do the authors explain the difference in CD44+/CD24- expression between INTR.20 and INTR.40 and between CONT.5 and CONT.15.
Table 2: how do the authors explain the difference in drug resistance observed among the different treatments? (Both CONT. and INTR.). Is there any relationship between the drug response and the level of CD44+/CD24- expression reported in the previous experiment?
Figure 8: if possible, add a positive control for the transwell migration (a purified chemokine) to give an estimation of the migration ability.
Page 20, line 571: which is the parameter determined? Cell migration or cell proliferation?
Figure 9 and 10: poor quality. Please improve the microscope images or, if not possible, show as example only a control and a treated, plus the quantification. In Figure 9 it is not possible to distinguish the migrated cells and the empty space.
The authors quantify the presence of the secreted VEGF in the medium of mammospheres and then concluded that this is responsible of the endothelial net-like structures. Although VEGF is surely a good candidate for the induction of the described ability, a direct evidence in the experiments presented is lacking. I suggest at least to add a treatment with a neutralizing specific VEGF antibody to verify the contribution of the latter in the mechanisms observed.
The authors should also give an explanation of the rationale underlining the use of different hypoxia treatments (continuous and intermittent) and discuss the findings they obtained comparing the two and within each one, the different dose-response they obtained.

Reviewer 2 ·

Basic reporting

This article meets the standard for publications.

1) The English used in the articles needs to proofread and edited by a language editor. At multiple places, the sentence and grammar used to make it difficult to understand.

2)The literature references and background context have been provided sufficiently.

3) Figure and tables have been structured appropriately.

4) hypothesis in the manuscript have been supported by relevant results.

Experimental design

The research presented in this manuscript is within the general scope of PeerJ journal. Overall the methods have explained, and the experiments have been designed to answer the question asked at each stage of the manuscript. Sufficient "n" has been used and the correct statistical test has been performed.

Validity of the findings

Authors use AnaeroGen Compact to create a hypoxic environment and make an assertion that it creates an environment of less than 1% oxygen. Have the authors confirmed then with some electronic or biochemical test? If not they should mention that AnaeroGen was used to create a Hypoxic environment.

For the INTR and CONT treatments used and motioned in the paper. How did the authors arrive at this conditions, what is the rationale for these specific conditions was it and trial and error method or was it based on from previous publications.

Authors need to show some evidence that during INTR or CONT incubation periods cells do experience the hypoxic condition. One possible way would be to Western blot for HIF immediately after hypoxia exposure. (The authors would have to be quick for collecting the cells has HIF1a has short half-life)

Authors use CD44+ positive bead selection. This could be a point of concern as in multiple cases the beads get bound to cells for a prolonged period of time and lead to stimulation of cells in an unintended way. Flow showing Enrichment of cd44+/cd25- after magnetic bead sort should be shown in the supplemental figure.

Migration of HUVEC and Elisa show that under hypoxia Cbc cells have a secretory phenotype. It would be informative to profile the cytokines in the media using multiplex bead-based assay like luminex.

Do authors think that these CSC between INT and CONT are differnet metabolically and doing assay like oxygen consumption, glucose metabolism could sheed light on further differnence.

Minor commets:-

In mammosphere culture protocol at line 146 authors mention that 2ml of media was topped off every 2 days till day seven. In that case, the total volume would be 16 ml for in a well volume of 16.8ml which could affect gas exchange. Was the approach used as authors think that mammosphere secrete factors essential for there growth and removing the media completely would hamper the growth?

At multiple places reference to figure/table is missing.

The images are missing scale bar and have and have 3D elevation in Figure 2 and 3. Please remove elevation its distracting. Also, mention the magnification at which images were taken.

Reviewer 3 ·

Basic reporting

The manuscript is written in clear and professional English language.
The introduction is comprehensively contextualizing the issues of the subject. The objectives of the work could be better defined at the end of the introduction.
Results- numeric data are showed inside parenthesis, for example: line 428 “ In contrast, the parental MCF-7 cells had only (1 ± 0.1%) CD44+ /CD24- expression and the unsorted mammospheres had (33.2 ± 430 3%) CD44+ /CD24- expression “ . It should be: “In contrast, the parental MCF-7 cells 429 had only 1 (± 0.1%) CD44+ /CD24- expression and the unsorted mammospheres had 33.2 (± 430 3%) CD44+ /CD24- expression.
Figures- The quality of the figures are in general good. I don’t like the 3D effects in figures 2 and 3. Please, improve the quality of pictures in figures 10 and 11.
Discussion- The first part of Discussion is repetitive of Introduction. CSC is well described in the introduction. In my opinion, it could begin in line 599.

Experimental design

The experimental approach is quite interesting and the obtained data are relevant to the better understanding of tumor cell biology. Figure 1 illustrates very well the experimental schedule.

Validity of the findings

Data are relevant and well discussed until CONTR10 and INTR30, when they observed increased in drug resistance, as well as the enhanced in capillary tubular structure formation. The effects of INTR30 and CONT10 are not adequately discussed.

·

Basic reporting

Basic reporting is redundant. Lacks the flow and transition, too lengthy.

Experimental design

Experimental designs are simple and easy to follow

Validity of the findings

no comment

Additional comments

1. The authors may want to discuss what is known about MCF7 phenotypes in 3D culture or hypoxia in general.
2. The results and discussion are extremely wordy and sometimes deviating from the objective of this paper. Figure 2: A, B, C and D legends are missing from the figure.
3. Figure 3: magnification of panel E and G do not look equivalent to A,B,C,D, F and H.
4. Figure 4: Flow cytometers dot plots. Gating for A,G and H have been moved. Needs to be exactly similar to the rest of the dot plots.
5. For better understanding of the figure, the % population of CD24-CD44+ should be written on the dot plot itself.
6. Figure 10 and 11: Poor representation of the images, dotted lines in the cell monolayer gap can make the images more clear.

---

## Round 0.2 · Minor Revisions

You will see that the reviewers appreciate your revisions. The only remaining problems relate to the language and grammar. The reviewer has provided a list of corrections, but this is not a comprehensive list and there are other sentences that need to be improved.

Reviewer 1 ·

Basic reporting

Although the quality of English has improved, I would recommend an extra check by a mother tongue, as several mispellings are still present.

Experimental design

The experiments are well designed and the results have been presented with the higher rate of details allowed by the authors' working system.

Validity of the findings

The discussion section has been improved with indications about the different hypoxia treatments.
Figure legends have been modified.

Additional comments

The quality of the manuscript has improved. I would like to thank the authors for the detailed response to my questions. Nevertheless, there are still few issues concerning punctuation and language that need to be addressed. Here are some points:
Page 2 line 51: “…throughout the body; including…” change “;” with “,”
Page 2 line 62: “The exact signaling pathways by which hypoxia activates its effects is complex..” change “is” with “are”
Page 2 line 65: change “&” with “and”
Page 3 line 75: “chronic hypoxic exposure is incubating cells in hypoxia up to several weeks”, change “is” with “is obtained by”
Page 3 line 89: “CSCs secretions help recruit endothelial..”, change in “…help to recruit…”
Page 11 line 382: “…but their number became sufficient only 3 days after which the cells were identified via FACS analysis…” this sentence is not completely clear. Three days after what? Sorting?
Page 12 line 419: “chemoresistance; MTT assay”, change “;” with “,”
Page 13 line 433: “…that mammosphere cultures are enriched with CSCs contributing to the higher drug resistance toward doxorubicin when compared to the parental MCF-7.” Please check the grammar of this sentence.
Page 13 line 461-462: “angiogenesis in breast cancer; quantitative assays for the main angiogenic factor; VEGF-A was performed” change “;” with “,”
Page 14 line 484-486: “These findings suggest that CSCsHYP secretionscomprised in CdM from CONT.5 and INTR.20 episodes significantly promot the migration of HUVECs.” Check this sentence for grammar and meaning.
Page 14 line 493: “To clarify the interaction between endothelial cells and CSCsHYP in tumor microenvironment; HUVECs were examined”, change “;” with “,”

Reviewer 2 ·

Basic reporting

The article has been revised and the current version is improved in terms of english and context.

Experimental design

Experiments are clear and the methods are reported in details.

Validity of the findings

Authors have addresed all the corncerns and the manuscript is now well crafted and acceptable.

---

## Round 0.3 · accepted · Accept

Many thanks for submitting your revised manuscript which has been improved over the previous versions.

#